# Grandparents’ Professional and Educational Activity: A Positive or Negative Impact on Relationships with Grandchildren?

**DOI:** 10.3390/ijerph20032248

**Published:** 2023-01-27

**Authors:** Dorota Kwiatkowska-Ciotucha, Alicja Grześkowiak, Urszula Załuska, Piotr Peternek

**Affiliations:** 1Department of Logistics, Wroclaw University of Economics and Business, 53-345 Wrocław, Poland; 2Department of Econometrics and Operational Research, Wroclaw University of Economics and Business, 53 345 Wrocław, Poland

**Keywords:** working grandparents, family relations, quantitative research, comparative analysis, statistical analysis of survey results, factor analysis, SEM

## Abstract

‘Baby boomers’ are the first generation whose representatives—both male and female—are, to a great extent, or plan to be, professionally active in their old age. Increased professional activity of this group influences family relations and the perception of the roles of grandmothers and grandfathers. This article attempts to assess the impact of grandparents’ professional and educational activity on relationships with grandchildren. The study relies on data from an international comparative survey conducted using the CAWI method in representative groups of Internet users from seven European countries and a proprietary questionnaire. The study was conducted in May 2022, and the total research sample included 3008 people. The analysis covered answers to the questions on the perceived impact of grandparents’ activity on the performance of family roles. To analyze the results, tests of the equality of means (*t*-test, ANOVA) were used to check for differences in assessments due to respondents’ characteristics. The use of exploratory factor analysis made it possible to distinguish two groups of factors—those having a positive impact on the relationship with grandchildren and those having a negative one. The SEM (structural equation modeling) model was used to find an answer to the question of which factor—positive or negative—has a greater impact on trust in grandparents as carers. The results of the conducted analyses indicated the existence of statistically significant differences in the perception of the role of grandmothers and grandfathers due to such characteristics of the respondents as age, gender, or country of residence. It is possible to conclude that positive perceptions of grandparents’ professional and educational activity encourage greater confidence in them in the context of performing family roles.

## 1. Introduction

Nowadays, 50-, 60- or even 70-year-olds, more frequently than previous generations, are likely to have their own career plans, want to be active, and want to develop themselves [1,2,3], and consequently—take care of their grandchildren to a lesser extent. Despite reaching their retirement age, they want to remain active in the labor market and participate to a greater extent in various forms of lifelong learning both for professional purposes and to pursue their own interests, hobbies, and dreams. At the same time, in the 21st century, they have an increasingly important role to play in raising the next generation, which results, among others, from longer life expectancy or the development of modern technologies that support people’s health despite old age. The need to take care of the younger generation is also connected with the increasing number of single parents or the globalization of work and, above all, the increase in the percentage of working women [4]. Many grandparents bridge the parenting gap by helping their grandchildren with school duties or advising on important life decisions. Grandparents can positively influence their grandchildren and can be role models for them [5]. Interestingly, the roles of grandparents are more and more often associated with both grandmothers and grandfathers, which is new compared to the previous associations connected with this role and attributing responsibilities mainly to grandmothers [6,7]. Changes in attitudes towards family roles described by many authors indicated, however, that women are still more heavily burdened with unpaid family care than men [8,9], which manifests itself, among others, in the reduced number of working women, and consequently in lower pension benefits for them in the future [10,11,12]. The evidence of negative relations between the daily care of grandchildren and employment is the most visible in countries with a familistic approach to care, i.e., countries characterized by relatively weak formal childcare and short effective parental leave (e.g., Greece, Italy, or Poland) [13].

Cross-sectional articles describing the results of research on the changing role of grandparents emphasize, among others, the great importance of cultural differences in the performance of this role and the need for in-depth research in this field [9]. Articles presenting an analysis of the results of research conducted in European countries in the area of grandparenting indicate an increasingly important role of grandparents in the upbringing of the next generations and changes in the performance of family roles and in the professional and educational activity of mature people, especially the representatives of the baby boomers’ generation [14]. An interesting view on this issue can be found in the results of research conducted in Hong Kong, where the diminishing role of intergenerational support in recent years has been indicated [15]. According to the research, the respondents are cautious about the role of grandparents as carers of younger generations, pointing to, among others, their insufficient knowledge of modern methods of upbringing.

The importance of grandparenting and the observed differences in attitudes towards the topic prompted the authors to design an international comparative study on the perception of the role of grandparents in contemporary societies in the context of their professional activity. The analysis of the results of the study designed in such a way allowed the authors to investigate the opinions of different groups on the involvement of grandparents in family life. Using the results of an international survey, an attempt was made, among others, to assess the perception of the impact of professional and educational activity of grandparents on relationships with their grandchildren. Three research questions were formulated for the purpose of the article:

The first research question: Are there any statistically significant differences in the assessment of the impact of professional and educational activity of grandparents on the relationship with grandchildren according to respondent characteristics such as age, gender, or country of residence? A professional activity is understood as remaining in any form on the labor market. An educational activity is any activity undertaken as part of lifelong learning in the form of formal, informal, or non-formal education.

The second research question: Is it possible to distinguish between a positive and negative impact of grandparents’ professional and educational activity on the relationship with grandchildren?

The third research question: Which of the consequences of grandparents’ professional and educational activity are more likely to translate into trust in them as carers?

## 2. Materials and Methods

The comparative research including representatives aged 18–65 from 7 European countries was conducted in May 2022 using computer-assisted Internet interviews (CAWI). The survey was commissioned by a professional public opinion polling institution. The sample was representative in each country in terms of characteristics such as gender, age (age groups), and place of residence (size). Additionally, the respondents taking part in the survey had different levels of education. The survey questionnaire was validated in a linguistic manner—it was translated by native speakers in both ways (from English to the national language and from the national language to English) and also evaluated in terms of understanding the intent of questions. The results of the validation indicated the correctness of the translation and understanding of the intent of the questions included in it. During the study, 3008 complete questionnaires were collected. The study involved at least 500 people from Finland, Germany, Greece, Poland, and Spain and at least 250 people from Belgium (Flanders) and the Netherlands. As for Belgium (Flanders) and the Netherlands, the reduced samples result from the inability to recruit such a large number of respondents due to the much less numerous CAWI panels for these countries than those previously mentioned. The research questionnaire concerned various aspects related to the broadly understood roles of grandmothers and grandfathers in the modern world. The questions focused, among others, on the perception of the role of grandmother and grandfather or the forms of childcare. The main subject of the study was the impact of grandparents’ professional and educational activity on family relationships, including relationships with grandchildren. In this article, respondents’ answers to 8 of the total 45 substantive questions in the survey were used for analysis.

The characteristics of the respondents in terms of the specific features included in the survey questionnaire are shown in Table 1. The selection of respondents’ characteristics was not random—the focus was on gender, age, and country of residence, that is, the characteristics which are identified in the literature as particularly differentiating people’s opinions [16,17]. It is worth noting that for the characteristic of the age, instead of equal age ranges, the division into generations adopted in the literature was used: baby boomers, generation X, generation Y, and generation Z [18,19,20]. The correctness of such a division is based on the assumption that differences resulting from respondents’ age may be caused by different attitudes towards work and family duties displayed by the representatives of different generations [21]. Because of this, the division into generations is substantively justified.

The distribution of characteristics presented in Table 1 is consistent with the structure of respondents in each country for the characteristics of gender and age. The number of the representatives of baby boomers and generation Z was lower than that of representatives of the other two generations due to the fact that the 18–65 age group consists of roughly half of the representatives of these generations. In contrast, sample sizes in specific countries were in line with the CAWI assumptions and the limitations of survey panel size in Belgium and The Netherlands.

The results presented in this article focus on respondents’ opinions on eight statements concerning the impact of grandparents’ professional and educational activity on relationships with grandchildren. Half of these statements have positive connotations—grandparents’ professional and educational activity influences relationships with their grandchildren in a positive manner. These statements were marked with symbols P1–P4, where:

P1: Professionally active grandparents are better role models for grandchildren than professionally inactive ones.

P2: Grandparents’ educational activities foster creating better relationships with grandchildren.

P3: Professional development of grandparents fosters creating better relationships with grandchildren.

P4: The power of influence of professionally active grandparents on grandchildren is greater than that of those who are professionally inactive (greater trust in active people, the validity of their knowledge, etc.).

Other statements have negative connotations—grandparents’ professional work and educational activity hinder relationships with their grandchildren. These statements are marked with symbols N1–N4, where:

N1: Grandmothers’ professional activity reduces the intensity of their contact with their grandchildren.

N2: Grandfathers’ professional activity reduces the intensity of their contact with their grandchildren.

N3: The grandparent–grandchild relationship is weaker in the case of grandparents who are professionally active compared to those who are not.

N4: The grandchildren of professionally active grandparents experience less emotional support from them than the grandchildren of professionally inactive ones.

In addition to the main set of statements P1–P4 and N1–N4, the analysis used—as a dependent variable—the opinions of respondents concerning trust in grandparents as carers expressed when assessing the ‘Trust’ statement worded in the following way:

Trust: Care provided by grandparents is more trustworthy than other forms of childcare.

In order to express opinions on all the statements, the respondents used a five-point Likert scale where 1 meant ‘I do not agree with the statement at all’ and 5—‘I completely agree with the statement’.

Various data analysis methods, appropriate to the content of specific questions, were used to find answers to the research questions posed in the article.

Tests of the equality of two means were used to check for differences in the assessment of the impact of grandparents’ professional and educational activity on the relationship with grandchildren depending on the characteristics of the respondent. The use of a particular method depended on the number of distinguished categories for the characteristics selected for analysis. For gender, an independent two-sample *t* test was applied. Tests of the equality of two means were preceded by Levene’s test for homogeneity of variances. When heterogeneity of variance was found, an alternative to the classical approach, the Welch *t*-test statistic, was applied. However, for the other two characteristics, i.e., generational group (generation) and country, a one-way analysis of variance was used. When ANOVA results showed significant differences, post hoc Tukey’s HSD tests for multiple comparisons were carried out to identify the pairs characterized by different means. The results of the post hoc tests are illustrated in figures, which makes it possible to clearly show the differences between the pairs of generational groups or countries being compared, even with a large number of combinations. In the figures, statistically significant differences between pairs of feature categories (nodes in the form of rectangles) are represented by arrows. The beginning of the arrow indicates the category of the characteristic with the statistically lower value of the test variable, whereas the head—the correspondingly statistically higher value of the mean score. The analyses take into account a threshold *p*-value at the level of 0.05, below which it was concluded that there are significant differences in the assessments of respondents characterized by different categories of characteristics.

Exploratory factor analysis (EFA) was used to investigate the structure of correlations between multiple variables. This involved determining whether several latent variables, which can explain the correlations between the primary observed variables to a large extent, can be identified. Before performing the EFA procedure, it was necessary to verify—with the use of the Bartlett test and the Kaiser–Meyer–Olkin (KMO) measure of the adequacy of the correlation matrix—whether the set of data justified its use [22]. In order to determine initial factors, principal components analysis was used. The number of factors making it possible to reduce dimensions was determined by the scree plot and the amount of explained variance. Orthogonal varimax rotation was used to obtain better interpretability of the factors [23].

Structural equation modeling (SEM) was used to assess the correlation between positive or negative perceptions of grandparents’ professional and educational activity and the perception of care provided by grandparents as more trustworthy than other forms of childcare (represented by the Trust variable). Such an approach allows for comprehensive testing of a complex structure, including one with latent variables [24]. Parameter estimation was carried out using the maximum likelihood method. The quality of the proposed model was evaluated on the basis of numerous criteria: factor loading values, parameter significance, and goodness-of-fit statistics specific to the SEM approach as root mean square error of approximation (RMSEA), the goodness of fit index (GFI), adjusted goodness of fit index (AGFI), comparative fit index (CFI), incremental fit index (IFI) and minimum discrepancy (CMIN/DF) [25,26].

Calculations were performed using the SPPS program, whereas structural equation modeling was carried out using SPSS AMOS ver. 28.0.0 (IBM SPSS, Chicago, IL, USA).

## 3. Results

In the beginning, the differentiation of respondents’ opinions when assessing the impact of grandparents’ professional and educational activity on relationships with grandchildren was analyzed, taking into account the characteristic of gender. The results are presented in Table 2.

Statistically significant differences were noted for all the negative statements (*p* < 0.000). In each of these statements, the mean scores given by male respondents were significantly higher than those given by women. It means that men, significantly more often than women, notice a negative impact of grandparents’ professional activity on relationships with their grandchildren, especially in terms of limited contact. According to men—and compared to opinions expressed by women—grandchildren of working grandparents receive less emotional support from them, and the relationship is not as strong as with grandparents who are professionally inactive. As far as positive statements are concerned, it was possible to note a significant difference of means only for the statement concerning the influence that professionally active grandparents have on their grandchildren. In this case, the mean score for men was also higher than that for women, but the difference in means was smaller than for the statements of a negative nature. Male respondents, more often than female ones, agreed with the opinion that working grandparents have a stronger influence on grandchildren than non-working ones, which probably results from greater up-to-date knowledge that working grandparents have and the greater trust that they gain.

Another characteristic analyzed was generation groups into which the respondents were classified based on their age. According to the state in 2022 and the birth year of the respondents, they were divided into the following groups: baby boomers (BB) 58–65 years old, generation X 42–57 years old, generation Y 26–41 years old, and generation Z 18–25 years old. The ANOVA results for the generation groups are shown in Table 3.

The results of the analysis of variance indicated that statistically significant differences between the opinions of representatives of different generations occurred for three questions with positive and two questions with negative connotations. The greatest diversity of opinions was noted for statement P2: *Grandparents’ educational activities foster creating better relationships with grandchildren* (*p* < 0.000). The respondents also expressed different opinions regarding grandmothers and grandfathers as role models for grandchildren and the impact of their professional development on their relationship with grandchildren. When it comes to negative statements, significant differences were noted for statement N3: *The grandparent–grandchild relationship is weaker in the case of grandparents who are professionally active compared to those who are not,* and N4: *The grandchildren of professionally active grandparents experience less emotional support from them than the grandchildren of professionally inactive ones*. A detailed analysis of the differentiation of opinions between the pairs of generation groups was carried out on the basis of the results of Tukey’s HSD tests. Statistically, significant differences are presented in Figure 1. The arrow indicates the occurrence of significant differences, with the arrowhead pointing towards the generation for which the mean of answers to a given statement was higher than for the generation to which the beginning of the arrow is located.

The largest number of significant differences between the pairs of generation groups was found in respondents’ answers to statement P2: *Grandparents’ educational activities foster creating better relationships with grandchildren*. It is possible to observe significantly higher mean values for the answers of respondents from generation X and baby boomers (BB) compared to those of respondents from generations Z and Y. It means that people from generation X and baby boomers appreciate more the educational activity of grandparents and its positive influence on relationships with their grandchildren. Similar differences can be found in the answers to statement P1: *Professionally active grandparents are better role models for grandchildren than professionally inactive ones,* and P3: *Professional development of grandparents fosters creating better relationships with grandchildren*. Significantly higher mean values were observed for the answers to statement P1 among the respondents from generation X and baby boomers compared to generation Y and baby boomers compared to generation Z. In the case of respondents’ answers to statement P3, significantly higher scores were given by the respondents from generation BB compared to the respondents from generations Y and Z. Thus, it is possible to note a more positive perception of grandparents’ professional activity and its impact on the relationship with their grandchildren among the representatives of baby boomers. However, the situation is quite different when analyzing negative statements concerning the impact of grandparents’ activity on relationships with grandchildren. As for statements N3: *The grandparent–grandchild relationship is weaker in the case of grandparents who are professionally active compared to those who are not* and N4: *The grandchildren of professionally active grandparents experience less emotional support from them than the grandchildren of professionally inactive ones,* there were significantly higher scores in the answers given by the respondents from generation Z compared to generation X. Thus, generation Z, compared to generation X, more often notices the negative impact of grandparents’ professional activity on the relationship with their grandchildren.

The last characteristic analyzed was the country of residence of the respondents. The results of the assessment of the differentiation of opinions due to this characteristic are presented in Table 4.

The ANOVA results indicated that for all the statements, both positive and negative, there were significant differences in the mean of the answers given by CAWI respondents living in analyzed European countries. For 5 out of 8 statements, the *p*-value was close to 0.000. The highest F-statistic values were noted for statement P1: *Professionally active grandparents are better role models for grandchildren than professionally inactive ones*, and also for statement P3: *Professional development of grandparents fosters creating better relationships with grandchildren*, P2: *Grandparents’ educational activities foster creating better relationships with grandchildren,* and N1: *Grandmothers’ professional activity reduces the intensity of their contact with their grandchildren*. In order to identify significant differences between pairs of specific countries, Tukey’s HSD tests were carried out again. A graphical representation of significant differences can be found in Figure 2, with the same symbols as in Figure 1. Arrows indicate significant differences between countries—the arrowhead is pointing towards the country for which the mean of answers to a given statement was higher.

An analysis of the respondents’ answers to the statements, broken down by country, reveals different assessments, including numerous significant differences between the opinions of respondents from different countries. It is worth noting that the mean value of answers given by the respondents from Greece is significantly higher and that these answers concern both positive and negative statements. It means that these respondents appreciate the professional and educational activity of grandparents but, at the same time, clearly indicate the negative impact of this activity on the relationship with grandchildren. The second country where the scores of answers were, on average, significantly higher was Poland, but here the significant difference applies to all positive statements, that is, perceiving grandparents as good carers of grandchildren, but, at the same time, noting that their activity leads to less frequent, although not worse, contact with grandchildren.

On the other hand, it is important to point to the countries where the respondents tended to give significantly lower scores, for example, Finland—for all statements except P4: *The power of influence of professionally active grandparents on grandchildren is greater than that of those who are professionally inactive* the mean values of answers were lower than of the answers given by the respondents from other countries, namely Greece and Poland. Thus, it can be concluded that the respondents from Finland, compared to the respondents from Poland and Greece, do not think that the professional activity of grandparents influences the intensity of contact with grandchildren (cf. N1 and N2). Moreover, they are less likely to say that the professional or educational activity of grandparents improves their relationship with grandchildren (cf. P2 and P3). Similar findings can be formulated for the respondents from the Netherlands. They also have significantly lower mean values for both positive and negative statements. Therefore, it can be concluded that they do not believe (compared to the respondents from other countries) that grandparents’ professional or educational activity helps to build relationships with grandchildren, but at the same time, they do not think that this activity has a negative impact on these relationships.

Opinions expressed by the respondents from Germany are also interesting. Their answers to the statement: *Professionally active grandparents are better role models for grandchildren than professionally inactive ones* appeared to have significantly higher mean scores than the answers of respondents from other countries, which means that they consider a working grandmother or grandfather to be a good role model.

Statements relating to the impact of grandparents’ professional and educational activity on the relationship with their grandchildren are of a specific nature—positive or negative. The second research question posed in the article concerns the possibility of grouping them in this context. Exploratory factor analysis was used, the results of which made it possible to determine whether statements of a specific nature group into a single factor. Kaiser–Meyer–Olkin test of sampling adequacy was conducted to check the validity of the use of factor analysis. The KMO above 0.8 (KMO = 0.816) indicates sampling adequacy [27]. Additionally, Bartlett’s test of sphericity (df = 28, *p* = 0.000) confirms that the variables are not orthogonal. Hence, the dimensionality reduction can be justified and can lead to meaningful results. Factors were extracted using principal components analysis. The first factor explains more than 40% of the variance and the second nearly 20%, giving a total of around 60%. The subsequent increments of variance explanation are slight, which can also be noted in the scree plot (not provided here). The scree plot layout, the degree of variance explanation, and also the substantive analysis of the statements covered by the study prompted the authors to take two factors into consideration [22,28]. The solution was subjected to varimax rotation with Kaiser normalization, and the results obtained can be found in Table 5.

The pattern of the loadings after rotation indicates that the variables can be related to factors. The variables connected with the positive impact of grandparents’ professional work and educational activity were classified into the second factor, whereas the variables connected with the negative influence were classified into the first one. The extraction of the two factors and the values of division quality measures associated with it indicate that the exploratory factor analysis gave results consistent with substantive expectations. The respondents expressing critical opinions on grandparents’ professional activity in the context of their relationship with their grandchildren gave similar answers to all negative statements. Conversely, those who appreciate grandparents’ professional and educational activity perceive their role as having more authority, more influence on their grandchildren, and ensuring better relationships.

The results of exploratory factor analysis identified two factors related to positive or negative attitudes towards professional and educational activity in the context of relationships with grandchildren. It is interesting to associate these constructs with the assessment of whether care provided by grandparents is more trustworthy than other forms of childcare (variable Trust). The methodological approach that makes it possible is the structural equation model, which can be used to identify relationships between latent and observable variables. The versatility and flexibility of SEM allow the links between different aspects of the phenomenon under consideration to be verified.

The proposed structural equation model describing the analyzed relationships and the results of the coefficient estimation is shown in Figure 3.

Table 6 presents the goodness-of-fit statistics for the structural equation model analyzed. The chi-square statistic is 1763.6 (*p* = 0.000), and the CMIN/DF measure is equal to 14.109. These are high values, which is not a desirable situation but occurs quite frequently in the case of large samples, limiting at the same time the usefulness of the assessment based on these measures [29,30,31]. Since in the study presented here N = 3008, conclusions from measures based on chi-square are not reliable. The evaluation of the model was mainly based on the values of RMSEA, GFI, AGFI, CFI, and IFI. The RMSEA value is 0.07, which is an acceptable level lower than the suggested threshold of 0.08 [32]. The other measures are normalized in the range of <0.1>, with values close to 1 indicating a very good fit of the model. There is no consensus among researchers on the cut-off point—the values that are most often mentioned in the literature are greater than 0.9 or greater than 0.95. GFI = 0.97 indicates a very good overall fit of the model. Other measures, i.e., AGFI, CFI, and IFI of 0.947, suggest a good fit. The quality measures allow the proposed model to be accepted.

Table 7 shows the results for the measurement part of the model, i.e., the standardized values of the factor loadings.

The results are quite high for all the items, ranging from 0.540 (for P4) to 0.753 (for N1). All factor loadings are positive. There are no standardized values below 0.5, which would suggest a weak correlation and prompt further verification of the model. A stronger correlation was observed for the Negative latent variable (0.674–0.753) rather than for the Positive one (0.540–0.692). Both factors can be considered to be well represented by the proposed observable variables.

The correlation between the factors is 0.42, indicating that they are not very strongly linearly correlated, do not duplicate information, and contribute separately to the dependent variable.

The results for the measurement part indicate that the model does not reject the assumed construct. Regression results for the impact of Positive and Negative latent variables on trust are included in Table 8.

Regression coefficients were used to make conclusions about the significance, direction, and strength of the impact of latent variables on the dependent variable that underwent analysis. Both predictors have a significant impact on the assessment of trust in grandparents as carers of their grandchildren (*p* < 0.001). The coefficients at both latent variables are positive, but standardized estimates indicate that the Positive factor is stronger. However, it should be noted that despite the significance of the impact, the standardized values suggest that it is not very strong, especially for the Negative factor (0.106), where it can be concluded that it is rather slight. Thus, it is possible to draw a general conclusion saying that positive perceptions of grandparents’ professional and educational activity foster greater confidence in them in the context of performing family roles and taking care of their grandchildren.

## 4. Discussion

The analysis of the results of the international study revealed statistically significant differences in respondents’ perceptions and opinions on grandparents’ professional and educational activity in the context of their relationship with their grandchildren. As far as we know, no analogous research has been conducted so far. A common focus of research in this area has been the influence of taking care of grandchildren on grandparents’ health, mood, and well-being [15,33,34,35,36]. Other research concentrated on the influence of grandparents’ care on the activity in the labor market of both grandparents and parents [11,13,37]. The differences found in our research were noted for all the analyzed characteristics, i.e., gender, age (represented by the respondent’s adherence to a given generation), and country of residence. The main conclusions from the analysis of differences in respondents’ opinions can be summarized as follows:-The most significant differentiation of opinions was noted for the characteristic of the country of residence. For this feature, statistically significant differences were observed for answers given to all eight analyzed statements. The country with the highest mean scores, regardless of the positive or negative nature of the statements, was Greece. Greeks gave higher scores to all the statements—both negative and positive—than respondents from any other country. High mean values were also noted in Poland—another country with a familistic approach towards care [13]. At the other extreme was Finland, where respondents showed great restraint in expressing their opinions. The scores of their answers were significantly lower for six out of eight statements compared to the answers given by Greeks, for four out of eight compared to the answers given by Polish respondents, and for three out of eight compared to the answers given by the respondents from Germany. The observed differences according to the country surveyed confirm the different perceptions of family roles related to the cultural dimensions. At this point, it is worth mentioning the masculine and feminine culture dimension (MAS) identified by G. Hofstede [38], which is mostly responsible for the differences in role perceptions.-An analysis of the results by age groups shows the greatest differences in the perceptions of grandparents’ professional and educational activity and its impact on the relationship with their grandchildren in the youngest generation represented in the study, generation Z. Statistically significant differences were noted in five out of eight statements, and they were mainly noted between generations Z and X or baby boomers. The results indicate that older generations, especially baby boomers, assess the impact of grandparents’ professional and educational activity on relationships with grandchildren in a positive way more frequently than the younger generations, whereas generation Z notices the negative impact of such activity more frequently than generation X.-The least significant differences were noted for the characteristic of gender. Interestingly, men tend to perceive a negative impact of grandparents’ professional activity on the relationship with grandchildren more frequently than women.

When referring to the second research question, it should be stated that the exploratory factor analysis revealed the existence of two factors that can be described as the positive and negative impact of grandparents’ professional and educational activity on the relationship with grandchildren. It is important to mention that the identification of such factors was in line with the substantive rationale and the verified statements. In their work, Buchanan and Rotkirch [4] indicated a positive influence of relationships between grandparents and grandchildren on mental health, resilience, and pro-social behavior. Falbo [39] found that stronger relationships with better-educated grandparents translated into better academic performance.

Structural equation modeling confirmed the existence of two latent factors with positive and negative connotations, both of which have a statistically significant impact on the degree of trust in grandparents as carers of grandchildren. The standardized values of the coefficients allow for the conclusion that the impact of the positive factor is stronger than that of the negative one. Thus, it can be concluded that a positive attitude towards grandparents’ professional and educational activity in relation to the relationship with their grandchildren translates into a greater degree of trust in them as carers than it was noted for a negative attitude. However, it is necessary to indicate that the strength of the impact, although statistically significant, is not extensive. The trend observed in European societies towards longer working lives, accompanied by lifelong learning activities, may have a positive impact on trust in grandparents as carers of their grandchildren.

SEM modeling made it possible to refer to the third research question—positive opinions on grandparents’ professional and educational activity translate into greater trust in them as carers. The conducted research has both strengths and limitations. The results presented in this article refer to an international comparative survey of CAWI panel respondents aged 18–65. The survey was conducted in seven countries of the European Union; therefore, it does not provide an overview of the situation for the countries of the entire European Union. It might be worth conducting a similar survey in other EU countries and comparing the results obtained.

The survey was designed and set in the Western world and does not directly translate to the non-Western reality, where the status of grandparents in society and the family is shaped differently.

Most research on relationships between grandparents and grandchildren focuses on posing questions directly to one or the other and identifying patterns within these groups. Our survey considers and analyses the views of a larger group of respondents (18–65-year-olds) in relation to the impact of grandparents’ professional and educational activity on relationships with grandchildren. The development and shaping of the general social attitude towards the caring roles performed by grandparents are very important in an era of ongoing demographic and cultural changes resulting in longer professional activity and the potential combination of work and taking care of grandchildren. On the one hand, longer professional activity limits grandparents’ ability to take care of their grandchildren, which means they have to work out a proper work–life balance. On the other hand, younger generations can benefit from relationships with grandparents who are still pursuing their careers, who keep acquiring new skills, and who are familiar with technological advances. What really matters is whether societies see this as an opportunity or as a disadvantage, which is what we tried to identify in our research, also taking into account gender, age, and country of residence.

## 5. Conclusions/Future Research

Arpino et al. [40], on the basis of data from the Survey of Health, Ageing, and Retirement in Europe, analyzed the correlation between subjective well-being and grandparental childcare, stating the general principle that they are positively associated with each other. They also indicated that the level of education of grandparents does not have a moderating effect on this relationship. In our study, the educational activity of grandparents is seen as a factor that fosters building better relationships with grandchildren, especially by the representatives of generation X and baby boomers. Further research might verify whether these associations are moderated by the level of education of grandparents. A similar study was carried out by Zamberletti et al. [41], who used data from the Multiscopo Family and Social Subjects survey conducted in Italy. In the study, they verified, using Multilevel multinomial logistic regression, the factors influencing the type of care provided by grandparents. These factors included characteristics such as gender, economic and health status, but also labor market status. It seems reasonable to compare the results of the Italian study with those carried out in other EU countries, taking into account not only professional but also educational activities. This would make it possible to directly determine the impact of these activities on the type and perhaps the quality of childcare.

We believe that the applied tool approach, which was presented in the article, can be used in research conducted in other countries, including developing ones. However, the conclusions resulting from the use of structural modeling (SEM) are not suitable for replication in developing countries, mainly due to cultural differences and differences in the level of development. The questions posed in the questionnaire are adapted to the situation in Europe, not in developing countries (e.g., in Asia or Africa), due to, for example, restricted professional activity of women, the territorial limitation of employment to major cities, etc.

## Figures and Tables

**Figure 1 ijerph-20-02248-f001:**
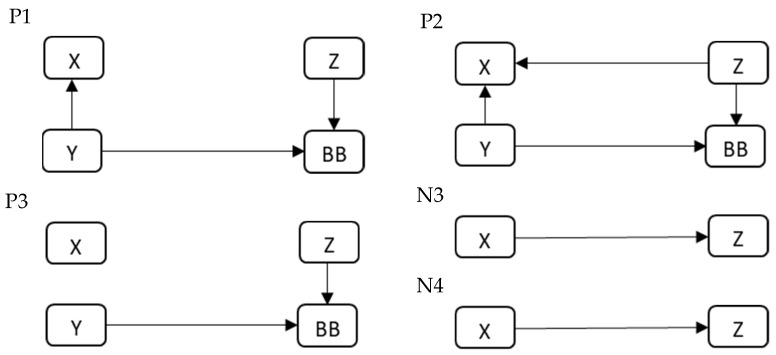
Visualization of significant differences in the answers to statements P1, P2, P3, N3, and N4 given by the respondents from different generations based on the post hoc tests.

**Figure 2 ijerph-20-02248-f002:**
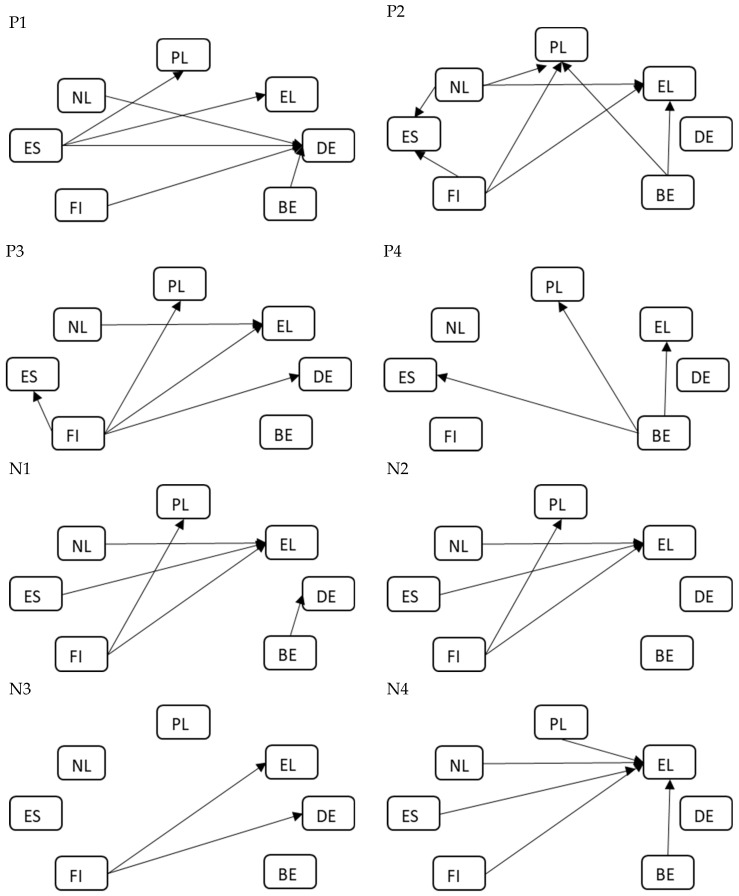
Visualization of significant differences in the answers to statements P1–P4 and N1–N4 given by the respondents from different countries based on the post hoc tests.

**Figure 3 ijerph-20-02248-f003:**
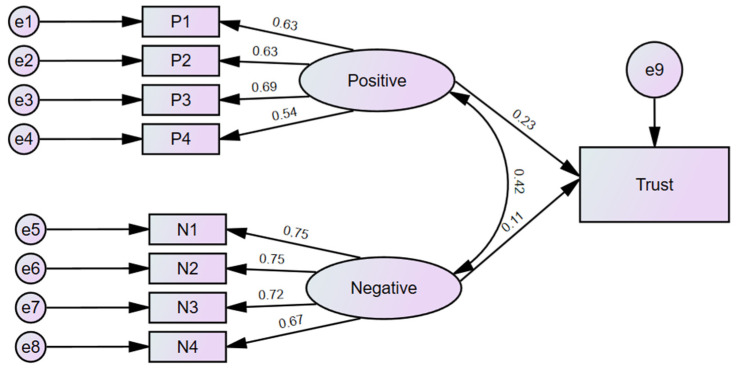
Structural equation model with standardized estimates.

**Table 1 ijerph-20-02248-t001:** A research sample—a structure according to selected characteristics (*N* = 3008).

Characteristic	Characteristic Categories	Percentage of Respondents
Gender	Female	49.8
	Male	50.2
Age (Generational groups)	Baby boomers (BB)	14.1
	Generation X (X)	36.7
	Generation Y (Y)	32.3
	Generation Z (Z)	16.9
Country	Belgium (BE)	8.3
	Finland (FI)	16.7
	Germany (DE)	16.7
	Greece (EL)	16.7
	Netherlands (NL)	8.3
	Poland (PL)	16.6
	Spain (ES)	16.7

**Table 2 ijerph-20-02248-t002:** Results of *t*-test for the grouping variable Gender.

No	Statement	t Statistic	*p*-Value
P1	Professionally active grandparents are better role models for grandchildren than professionally inactive ones	−0.347	0.729
P2	Grandparents’ educational activities foster creating better relationships with grandchildren	1.896	0.058
P3	Professional development of grandparents fosters creating better relationships with grandchildren	1.459	0.145
P4	The power of influence of professionally active grandparents on grandchildren is greater than that of those who are professionally inactive (greater trust in active people, the validity of their knowledge, etc.)	−2.511	0.012
N1	Grandmothers’ professional activity reduces the intensity of their contact with their grandchildren	−6.095	0.000
N2	Grandfathers’ professional activity reduces the intensity of their contact with their grandchildren	−6.515	0.000
N3	The grandparent–grandchild relationship is weaker in the case of grandparents who are professionally active compared to those who are not	−5.085	0.000
N4	The grandchildren of professionally active grandparents experience less emotional support from them than the grandchildren of professionally inactive ones	−5.224	0.000

**Table 3 ijerph-20-02248-t003:** Results of ANOVA for the grouping variable Generation.

No	Statement	F-Statistic	*p*-Value
P1	Professionally active grandparents are better role models for grandchildren than professionally inactive ones	5.368	0.001
P2	Grandparents’ educational activities foster creating better relationships with grandchildren	10.275	0.000
P3	Professional development of grandparents fosters creating better relationships with grandchildren	5.834	0.001
P4	The power of influence of professionally active grandparents on grandchildren is greater than that of those who are professionally inactive (greater trust in active people, the validity of their knowledge, etc.)	1.241	0.293
N1	Grandmothers’ professional activity reduces the intensity of their contact with their grandchildren	0.414	0.743
N2	Grandfathers’ professional activity reduces the intensity of their contact with their grandchildren	1.105	0.346
N3	The grandparent–grandchild relationship is weaker in the case of grandparents who are professionally active compared to those who are not	3.051	0.027
N4	The grandchildren of professionally active grandparents experience less emotional support from them than the grandchildren of professionally inactive ones	4.132	0.006

**Table 4 ijerph-20-02248-t004:** Results of ANOVA for the grouping variable Country.

No	Statement	F-Statistic	*p*-Value
P1	Professionally active grandparents are better role models for grandchildren than professionally inactive ones	8.720	0.000
P2	Grandparents’ educational activities foster creating better relationships with grandchildren	6.752	0.000
P3	Professional development of grandparents fosters creating better relationships with grandchildren	6.939	0.000
P4	The power of influence of professionally active grandparents on grandchildren is greater than that of those who are professionally inactive (greater trust in active people, the validity of their knowledge, etc.)	3.873	0.001
N1	Grandmothers’ professional activity reduces the intensity of their contact with their grandchildren	6.110	0.000
N2	Grandfathers’ professional activity reduces the intensity of their contact with their grandchildren	4.948	0.000
N3	The grandparent–grandchild relationship is weaker in the case of grandparents who are professionally active compared to those who are not	2.987	0.007
N4	The grandchildren of professionally active grandparents experience less emotional support from them than the grandchildren of professionally inactive ones	4.025	0.001

**Table 5 ijerph-20-02248-t005:** Factor analysis results—factor loadings after varimax rotation with Kaiser normalization.

No	Variable (Statement)	Loadings	
		Factor 1	Factor 2
P1	Professionally active grandparents are better role models for grandchildren than professionally inactive ones	0.146	0.732
P2	Grandparents’ educational activities foster creating better relationships with grandchildren	0.047	0.761
P3	Professional development of grandparents fosters creating better relationships with grandchildren	0.056	0.809
P4	The power of influence of professionally active grandparents on grandchildren is greater than that of those who are professionally inactive (greater trust in active people, the validity of their knowledge, etc.)	0.376	0.547
N1	Grandmothers’ professional activity reduces the intensity of their contact with their grandchildren	0.807	0.114
N2	Grandfathers’ professional activity reduces the intensity of their contact with their grandchildren	0.786	0.155
N3	The grandparent–grandchild relationship is weaker in the case of grandparents who are professionally active compared to those who are not	0.797	0.102
N4	The grandchildren of professionally active grandparents experience less emotional support from them than the grandchildren of professionally inactive ones	0.764	0.112

**Table 6 ijerph-20-02248-t006:** Structural equation model—characteristics and quality measures.

Measure	Value
Chi-square	389.989
d.f.	25
*p*	0.000
CMIN/DF (minimum discrepancy)	15.600
RMSEA (root mean square error of approximation)	0.070
GFI (goodness of fit index)	0.970
AGFI (adjusted goodness of fit index)	0.947
CFI (comparative fit index)	0.947
IFI (incremental fit index)	0.947

**Table 7 ijerph-20-02248-t007:** Factor loadings.

Latent Variable	Item	Standardized Factor Loading
Positive	P1	0.630
	P2	0.625
	P3	0.692
	P4	0.540
Negative	N1	0.753
	N2	0.750
	N3	0.719
	N4	0.674

**Table 8 ijerph-20-02248-t008:** Regression results: parameter estimates, standard errors, and standardized estimates.

Relation	Parameter Estimate	Standard Error	Standardized Estimate
Trust <--- Positive	0.489 ***	0.054	0.227
Trust <--- Negative	0.167 ***	0.036	0.106

*** *p* < 0.001.

## Data Availability

The dataset presented in the study is available upon request from the corresponding author.

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
