# Peer review of "Grandparents’ Professional and Educational Activity: A Positive or Negative Impact on Relationships with Grandchildren?"

_ijerph, 2023, doi:10.3390/ijerph20032248_

Round 1

Reviewer 1 Report

A great deal of present-day empirical research is centered on child/parents relationships. In contrast, child/grandparents relationships have not been properly explored. However, an increase in longevity contributes to the growth of grannies and granddads involved in the life of their grandchildren to this or that extent. According to the family systems theory, there are substructures within every family that form dyadic relationships between generations: parent-child, grandparent-grandson. Every kind of relationships has its special features affecting each family member in a different way and creating specific family milieu which becomes a determinant in the child’s development. The relevance of the topic resides in the fact that harmonious child-grandparental relationships can be a condition for social support and satisfaction in families. Besides, grandparental support can act as a substantial resource for middle generation and a source of effective social interaction for grandchildren.

 Questions and comments:

1.  Figures 1 and 2 are poorly informative.

2.  What do authors mean by formative (educational) activities?

3.  Since different generations participated in the survey the results are the following: younger generation evaluated their attitude to grandparents, while the elder one – their role of being grannies and granddads, which the authors did not take into consideration in their research.

Author Response

Dear Sir or Madam,

Thank you for all your comments and remarks. We carefully analyzed all of them. Below you will find our responses as well as suggested changes.

 Questions and comments:

  1. Figures 1 and 2 are poorly informative.

Thank you for bringing this to our attention. However, for post hoc tests for many of the characteristic categories there is no good tabular presentation of the results. Therefore, we decided to present them in our own graphical way using well-known and simple graphical elements such as an arrow and a text box with the name of the category. In the article, in lines 167-173, we suggested the following interpretation:

„The results of the post hoc tests are illustrated in figures, which makes it possible to clearly show the differences between the pairs of generational groups or countries being compared, even with a large number of combinations. In the figures, statistically significant differences between pairs of feature categories (nodes in the form of rectangles) are represented by arrows. The beginning of the arrow indicates the category of the characteristic with the statistically lower value of the test variable, whereas the head - the correspondingly statistically higher value of the mean score.”

  1. What do authors mean by formative (educational) activities?

By educational activities, we mean participation in formal, non-formal and informal forms of lifelong learning.

  1. Since different generations participated in the survey the results are the following: younger generation evaluated their attitude to grandparents, while the elder one – their role of being grannies and granddads, which the authors did not take into consideration in their research.

Thank you for bringing this to our attention. The described risks occur in all social research. The main objective of the survey conducted was to assess the perception of the role of grandmothers and grandfathers in the contemporary world. The survey was provided with a preamble outlining our expectations. We are aware that in social research, the respondent answers questions on base of his or her own experiences. In our survey, these differences were taken into account in terms of generational differences and were included in the research conclusions. We would like to emphasise that the age of the responded understood as a factor qualifying him or her to generational groups was identified in the results of our analyses as the main characteristic that differentiates the perception of the role of grandmothers and grandfathers.

Reviewer 2 Report

ijerph-2095284-peer-review-v1

It is a very interesting study, which evaluates the impact of the professional and educational activity of the grandparents in their relations with their grandchildren.

In order for the manuscript to be considered as fulfilling the list of essential points that must be described in the observational research publication.

It is necessary to restructure the abstract.

Define the variables of professional and educational activity.

Restructure the discussion section and mention the usefulness of the study.

Observations for the authors

In the summary section

What does SEM mean? Whenever an abbreviation is used for the first time, it must be defined.

The abstract section should present the key results of the study.

In the summary, the conclusion of the study is missing, and the objective of the study was fulfilled or not.

In the material and methods section

What was the percentage of incomplete questionnaires?

What was the reason for including only developed countries?

What was the total survey?

This survey was validated.

If it was validated, what were the results of the validation?

How was it verified that the correct person was answering the survey?

You calculated the size of the sample considering gender, the division into generations, and the country surveyed so that it was representative.

How was professional and educational activity defined?

In the discussion section

Rather what was presented in the discussion section is the repetition of the results.

In the discussion section they have to compare their own conclusions with those of other researchers, always speculate and theorize with common sense and logic about the general aspects of the conclusions, comment on anomalous findings, give them the most coherent explanation possible or simply state that this is true. certain. what was observed, although at the moment there is no possible explanation.

Mention the strengths and weaknesses of the study.

Do you think this model can be replicated in poor or developing countries, where the sociocultural context is very different?

Argue the reasons for the differences observed according to the country surveyed.

What are the implications of your results?

Author Response

Dear Sir or Madam,

Thank you for all your comments and remarks. We carefully analyzed all of them. Below you will find our responses as well as suggested changes.

Observations for the authors

In the summary section

What does SEM mean? Whenever an abbreviation is used for the first time, it must be defined.

The abstract section should present the key results of the study.

In the summary, the conclusion of the study is missing, and the objective of the study was fulfilled or not.

Thank you for your suggestions. We proposed such a form of abstract due to formal restrictions on the maximum number of words in this section of the article. The current form is as follows:

Abstract: ‘Baby boomers’ is the first generation whose representatives – both male and female - are, to a great extent, or plan to be, professionally active in their old age. Increased professional activity of this group influences family relations and the perception of the roles of grandmothers and grandfathers. This article attempts to assess the impact of grandparents’ professional and educational activity on relationships with grandchildren. The study relies on data from an international comparative survey conducted using the CAWI method in representative groups of Internet users from 7 European countries and a proprietary questionnaire. The study was conducted in May 2022, and the total research sample included 3,008 people. The analysis covered answers to the questions on the perceived impact of grandparents’ activity on the performance of family roles. To analyse the results, tests of the equality of means (t-test, ANOVA) were used to check for differences in assessments due to respondents’ characteristics. The use of exploratory factor analysis made it possible to distinguish two groups of factors – those having a positive impact on the relationship with grandchildren, and those having a negative one. The SEM (structural equation modelling) model was used to find an answer to the question of which factor - positive or negative – has a greater impact on trust in grandparents as carers. The results of the conducted analyses indicated the existence of statistically significant differences in the perception of the role of grandmothers and grandfathers due to such characteristics of the respondents as age, gender or country of residence.  It is possible to conclude that positive perceptions of grandparents’ professional and educational activity encourage greater confidence in them in the context of performing family roles

In the material and methods section

What was the percentage of incomplete questionnaires?

The survey was commissioned to be conducted by a professional public opinion polling institution. In line with our requirements, the total sample size should include at least 3,000 people according to the structure indicated in the article in lines 89 - 91. The sample was to be representative in each country in terms of characteristics such as gender, age (age groups) and place of residence (its size). Additionally, the respondents taking part in the survey were supposed to have different levels of education. The results obtained included only fully completed questionnaires. In line 89 we included some additional information:  

“The survey was commissioned to a professional public opinion polling institution. The sample was representative in each country in terms of characteristics such as gender, age (age groups) and place of residence (size). Additionally, the respondents taking part in the survey had different levels of education.”

What was the reason for including only developed countries?

The survey covered 7 European countries (see the Abstract section, lines 19 - 20). According to the OECD classification, all European countries belong to the group of developed ones. Therefore, all countries included in the study belong to this group.

What was the total survey?

The total size of the sample corresponds to the number of questionnaires collected, that is, 3,008 respondents (see line 89 in the article). A questionnaire consisted of 45 substantive questions, of which the article presents the results for 8 (see lines 99 – 100 in the article).

This survey was validated.

The survey questionnaire was validated in a linguistic manner – it was translated by native speakers in both ways (from English to the national language and from the national language to English) and also evaluated in terms of understanding the intent of questions.

If it was validated, what were the results of the validation?

The results of the validation indicated the correctness of the translation and understanding of the intent of the questions included in it.

How was it verified that the correct person was answering the survey?

The respondents taking part in the survey were panellists cooperating with the professional public opinion polling institution selected in line with the terms and conditions of the contract.

According to our requirements, the total sample size should consist of at least 3,000 people (see lines 89 – 91 in the article). The sample was to be representative in each country in terms of characteristics such as gender, age (age groups) and place of residence (size). Additionally, the respondents participating in the survey had different levels of education.

You calculated the size of the sample considering gender, the division into generations, and the country surveyed so that it was representative.

The screening questions included in the questionnaire ensure the representativeness of the sample in terms of the indicated characteristics of the respondents.

How was professional and educational activity defined?

By professional activity we meant remaining active in the labour market in any way. By educational activity we meant any activity undertaken as part of Lifelong Learning in the form of formal, informal or non-formal education.

In the discussion section

Rather what was presented in the discussion section is the repetition of the results.

In the discussion section they have to compare their own conclusions with those of other researchers, always speculate and theorize with common sense and logic about the general aspects of the conclusions, comment on anomalous findings, give them the most coherent explanation possible or simply state that this is true. certain. what was observed, although at the moment there is no possible explanation.

To our best knowledge, the relationship between professional and educational activity and the general social perception of the role of grandparents as carers for grandchildren has not yet been analysed in detail. The literature focuses on the effects of caregiving, both for grandparents and grandchildren. Effects on health, well-being, changes in labour supply, social behaviour and educational outcomes, among others, have been analysed. In the discussion, we included papers referring to this kind of effects and, in line with the reviewer’s suggestions, we have presented a broader context and literature references on relationships between grandparents and grandchildren.

Mention the strengths and weaknesses of the study.

Thank you for bringing this to our attention. The relevant information has been introduced in the article in the discussion section starting from line 463.

The conducted research has both strengths and limitations. The results presented in this article refer to an international comparative survey of CAWI panel respondents aged 18-65. The survey was conducted in 7 countries of the European Union; therefore, it does not provide an overview of the situation for the countries of the entire European Union. It might be worth conducting a similar survey in other EU countries and compare the results obtained.

The survey was designed and set in the Western world and does not directly translate to the non-Western reality, where the status of grandparents in society and the family is shaped differently.

Most research on relationships between grandparents and grandchildren focuses on posing questions directly to one or the other and identifying patterns within these groups. Our survey considers and analyses the views of a larger group of respondents (18-65 year olds) in relation to the impact of grandparents’ professional and educational activity on relationships with grandchildren. The development and shaping of the general social attitude towards the caring roles performed by grandparents is very important in an era of ongoing demographic and cultural changes resulting in longer professional activity and the potential combination of work and taking care of grandchildren. On the one hand, longer professional activity limits grandparents’ ability to take care of their grandchildren, which means they have to work out proper work life balance. On the other hand, younger generations can benefit from relationships with grandparents who are still pursuing their careers, who keep acquiring new skills and who are familiar with technological advances. What really matters is whether societies see this as an opportunity or as a disadvantage, which is what we tried to identify in our research, also taking into account gender, age and country of residence.

Do you think this model can be replicated in poor or developing countries, where the sociocultural context is very different?

We believe that the applied tool approach, which was presented in the article, can be used in research conducted in other countries, including developing ones. However, the conclusions resulting from the use of structural modelling (SEM) are not suitable for replication in developing countries mainly due to cultural differences and differences in the level of development. The questions posed in the questionnaire are adapted to the situation in Europe, not in developing countries (e.g. in Asia or Africa), due to, for example, restricted professional activity of women, the territorial limitation of employment to major cities, etc.

Argue the reasons for the differences observed according to the country surveyed.

What are the implications of your results?

Thank you for bringing this to our attention. Relevant literature references have been added to the discussion section.

Round 2

Reviewer 2 Report

Currently, the manuscript has improved significantly.

The authors have resolved all comments and observations, but some of the explanations must also be written in the manuscript.

The explanation of the validation of the questionnaire must be included in the manuscript.

Operational definitions of a professional and educational activities must be included in the manuscript.

Since one of the characteristics of the scientific method is that research can be replicated, I suggest that it is essential to include in the manuscript an explanation of whether the proposed model can be replicated in poor or developing countries.

Author Response

A version of the article with additional explanations is attached. Changes are marked in blue. In addition, due to the change in the name of the ministry sponsoring the research, we have made a correction in the Funding section. We also marked this change in blue.